# Monitoring Occupational Noise Exposure in Firefighters Using the Apple Watch

**DOI:** 10.3390/ijerph20032315

**Published:** 2023-01-28

**Authors:** Erin C. Williams, Yiran Ma, Daniela M. Loo, Natasha Schaefer Solle, Barbara Millet, Kristine Harris, Hillary A. Snapp, Suhrud M. Rajguru

**Affiliations:** 1Department of Otolaryngology, University of Miami, Miami, FL 33136, USA; 2Department of Biomedical Engineering, University of Miami, Coral Gables, FL 33136, USA; 3Department of Medicine, University of Miami, Miami, FL 33136, USA; 4Department of Interactive Media, University of Miami, Coral Gables, FL 33146, USA; 5RestorEar Devices LLC, Bozeman, MT 59715, USA

**Keywords:** noise-induced hearing loss, firefighters, occupational noise, Apple Watch, noise exposure

## Abstract

Occupational noise exposure and hearing loss are prominent in the fire service. Firefighters are routinely exposed to hazardous levels of noise arising from the tools and equipment they use, from sirens and alarm tones to the emergency response vehicles they drive. The present study utilized the Apple Watch to continuously measure environmental noise levels for on-duty firefighters. Participants included 15 firefighters from the metropolitan South Florida area, and 25 adult non-firefighter control subjects. Firefighters were recruited from a variety of roles across two stations to ensure noise exposure profiles were appropriately representative of exposures in the fire service. All participants wore an Apple Watch for up to three separate 24 h shifts and completed a post-shift survey self-reporting on perceived exposures over the 24 h study period. Cumulative exposures were calculated for each shift and noise dose was calculated relative to the NIOSH recommended exposure limit of 85 dBA as an 8 h time-weighted average. The maximum dBA recorded on the Apple Watches was statistically significant between groups, with firefighters experiencing a median of 87.79 dBA and controls a median of 77.27 dBA. Estimated Exposure Time at 85 dBA (EET-85) values were significantly higher for firefighters when compared to controls: 3.97 h (range: 1.20–14.7 h) versus 0.42 h (range: 0.05–8.21 h). Only 2 of 16 firefighters reported the use of hearing protection devices during their shifts. Overall, our results highlight the utility of a commonly used personal device to quantify noise exposure in an occupationally at-risk group.

## 1. Introduction

Occupational noise-induced hearing loss (NIHL) is a significant public health issue worldwide [1], with nearly one third of all cases of hearing loss attributed to dangerous noise exposure [2]. In the United States alone, it is estimated that approximately 22 million workers are exposed to hazardous noise at work [3]. In addition to irreversible hearing loss, NIHL has been shown to affect quality of life through impaired social interactions, occupational abilities [4], sleep disorders, cardiovascular disease, depression, and cognitive decline [5,6]. These consequences are particularly dire for firefighters [7,8], who depend on their hearing to perform essential job functions such as localizing sound in a rescue search, responding to radio communications, and hearing personal alert safety system alarms [9].

In the United States alone, there are approximately 1,080,800 career and volunteer firefighters according to available estimates [9,10]. While the potential for some occupational hazards is obvious, noise is ubiquitous in the environment and is a leading cause of acquired NIHL in this group, particularly over the span of a long career [11,12]. Both sound level intensity and duration of exposure contribute to NIHL, which develops slowly over time [13]. At present, the NIOSH recommended exposure limit (REL) for noise is 85 A-weighted decibels (dBA) as an 8 h time-weighted average (TWA) using a 3 dB exchange rate. Exposures at or above this level are considered hazardous [14], and are unfortunately experienced by firefighters routinely via sirens and alarm tones, water pumps, saws, emergency response vehicles, and from other equipment or machinery that generates excessive noise [15].

Previous work has characterized noise sources and hazardous noise levels for various firefighting activities in this group [8,12,15,16,17,18,19], typically in the form of short-term averaging or maximum sounds experienced over short intervals, as measured with sound level meters and dosimeters. These devices are accurate and often specialized, which can make them expensive. Many of these devices are bulky and not feasible for firefighters to safely carry around on their person. In recent years, several studies have demonstrated the utility of smartphones to measure noise [20,21,22,23]. Unfortunately, even these devices are variable in quality and depend highly on the location and sensitivity of the microphone(s), as well as the application used to collect data [24]. As a result, these have limited applications for monitoring sound level, duration, and frequency of exposure to hazardous noise in an active-duty scenario. The advent of smartwatches has provided an elegant solution to the limitations noted above, especially since they can be unobtrusively worn on the person by firefighters during their shifts. Not only have smart-watches become quite prevalent for personal use, they are also capable of providing a user-friendly and sufficiently accurate alternative to traditional sound level meters for continuous noise monitoring [25,26,27]. The small footprint of these devices also enables them to be worn while on duty without compromising safety.

In this work, the primary aim was to characterize and quantify noise exposure routinely experienced by firefighters. We also sought to investigate the feasibility of using the Apple Watch for continuous noise monitoring in this group.

## 2. Methods

The study was approved by the University of Miami’s Institutional Review Board (#20200222). Informed consent was obtained from all subjects involved in the study.

### 2.1. Study Design and Participants

Adult firefighters (*n* = 15) employed at a single station in the metropolitan South Florida area were recruited from a variety of roles in the fire service to ensure noise profiles were representative of routine occupational exposures. A control group (*n* = 25) of non-firefighter controls were recruited from the general population. Following the written informed consent process, all participants wore an Apple Watch Series 4 (Apple Inc., Cupertino, CA, USA) equipped with an internal noise meter for up to 3 separate 24 h shifts and completed a post-shift survey detailing exposures experienced during each shift. Participant demographics and relevant medical and hearing health histories were also collected. Each watch was paired with a research iPhone (Apple Inc., Cupertino, CA, USA) running iOS (Version 13 or higher, Apple Inc., Cupertino, CA, USA) for data storage. All other features were deactivated from the watch and no personal data were collected or stored from the watch during the study period. Study data were collected and managed using REDCap electronic data capture tools hosted at the University of Miami Miller School of Medicine [28,29]. REDCap (Research Electronic Data Capture) is a secure, web-based software platform designed to support data capture for research studies, providing (1) an intuitive interface for validated data capture; (2) audit trails for tracking data manipulation and export procedures; (3) automated export procedures for seamless data downloads to common statistical packages; and (4) procedures for data integration and interoperability with external sources.

### 2.2. Data Analysis

Noise data collected using the Apple Watches were extracted from the Health App and exported into MATLAB for further analysis. The data reported were averaged over each half hour—providing 48 data points over 24 h periods (or one shift). It was not feasible to download more granular data from the Apple Watch/Health application. Equivalent Exposure Time at 85 dB (EET-85) was calculated as $t_{eq}=\left(0.5\;h\right)*2^{\left(\frac{x-85}{3}\right)}$, and Estimated Exposure at 8 h (EA-8hr) was calculated as $dBA_{ave}=85+3*log_{2}\left(\frac{t_{total}}{8\;h}\right)$ for each participant and each shift completed. Using a custom MATLAB script, these values (EET-85 and EA-8hr) were compared to NIOSH standards for recommended exposure limits (REL) (i.e., 85 dBA over 8 h) [14].

All statistical analyses, including the Shapiro–Wilk test for normality, Pearson correlation, Welch’s *t*-test, and accompanying figures were generated in R Studio and are detailed along with the results.

## 3. Results

### 3.1. Participant Characteristics and Demographics

Fifteen firefighters and twenty-five controls were enrolled in the study (see Table 1). A majority of participants self-identified as White, with about half (53.3%) of the firefighters self-identifying as Hispanic compared to 64% of controls. The median age was 33 (SD: 10.64) and 37 (SD: 10.48) for firefighters and controls, respectively. Gender distribution was primarily male (80%) in the firefighter population, but relatively balanced in controls at 56% male and 44% female. The majority of firefighters had completed some college or technical school (66.7%), while controls were primarily college graduates (72%). All firefighters were career firefighters employed at a fire station in the greater South Florida metropolitan area. No participants in either cohort reported secondary employment, and none identified as a veteran. The control cohort was highly variable in occupation, though none were firefighters or employed in an industry considered at-risk for hazardous noise exposure.

### 3.2. Hearing Health and Pre-Existing Comorbidities at the Time of Enrollment

Sixty percent of firefighters reported having received a previous hearing evaluation, citing routine check-up as the only indication for examination. Among those who reported auditory or vestibular complaints in the past (27%), pediatric recurrent ear infections were most commonly noted (50%), with only one instance of tinnitus and one instance of dizziness/vertigo recorded. Thirteen (86.7%) had no history of dizziness or vertigo. Comparatively, 72% of controls reported that they had never seen a doctor or healthcare provider for a hearing evaluation. The remaining seven participants (28%) reported a routine check-up as the primary reason for evaluation, though legitimate otologic complaints, including tinnitus, hearing changes, dizziness, and disequilibrium were also recorded for one participant each. Eighty percent of controls had no history of auditory or vestibular complaints in the past, but the remaining participants (20%) noted tinnitus and pediatric ear infections (8% reported incidence of each symptom, respectively), and hearing loss, previous ear injections, and dizziness (4% reported incidence of each symptom, respectively). Ninety-two percent of controls had never been diagnosed with dizziness or vertigo. One control participant had a history of ear surgery. None of the participants in either group noted the use of hearing devices, including hearing aids, personal amplifiers, or cochlear implants in either group.

Other comorbid conditions polled, including cardiac disease and diabetes, were absent in our cohort, with the exception of one firefighter who reported a history of cardiac disease. Three (20%) firefighters noted previous head trauma or concussion. Similarly, four controls (16%) reported previous head trauma or concussion. All but one firefighter reported having health insurance, with a majority covered by (potentially additional) private insurance (12/15 [80%]). All but two control participants reported health insurance coverage, primarily private (68%), but also state sponsored (4.3%) and single service (4/25). Three (13%) participants were unsure of their health insurance coverage.

Firefighters reported having experienced hearing changes following a duty-related incident, with 26.7% experiencing tinnitus and 6.7% experiencing changes in hearing or muffled hearing (see Table 2). A majority (60%) also noted tinnitus up to 99% of the time. Most firefighters had never experienced imbalance or disorientation following noise exposure. Notably, no firefighters self-reported ever being exposed to hazardous noise. Survey responses ranged from rarely (1–24% of the time) to sometimes (50–74% of the time) being exposed to hazardous noise. Controls almost seldom reported the above symptoms and generally felt that they were not often exposed to hazardous noise, though 48% noted experiencing ringing or buzzing in their ears.

### 3.3. Self-Perceived Risk for Negative Hearing or Vestibular Outcomes

Over half (60%) of firefighters reported that they were not generally concerned about their hearing, with only five (33%) noting concern about their hearing in both ears (Table 3). Controls were even less concerned about their hearing, with 84% reporting that they were not concerned at all. As with the firefighters, a small number (12%) of controls expressed that they were concerned about hearing in both ears (Table 3). Accordingly, firefighters self-reported that they were largely non-compliant in wearing hearing protection (Table 2), with 70% reporting that they never or rarely wore it during hazardous noise exposure on shift. Firefighters did note the occasional use of hearing protection outside of work when exposed to hazardous noise. Additionally, 47% of firefighters reported that their employer had recommended the use of personal protective equipment (PPE), though they had largely (87%) not received information about hearing loss. Unsurprisingly, controls also reported that they did not wear hearing protection during hazardous noise at work, though it is unclear whether this was due to a general lack of noise exposure or non-compliance (Table 2). Outside of work, 72% never wore hearing protection, 8% rarely, and 4% each for occasionally, sometimes, and frequently. PPE was not recommended by work for control participants, and a majority (76%) reported that they had never received information about noise-induced hearing loss.

### 3.4. Hazardous Noise Exposure

The firefighter cohort was mostly comprised of firefighters/paramedics/EMTs (66.7%) employed for an average of 11.6 years (Table 4). Assigned units included EMS trucks (28.9%), fire engines (52.6%), or other (18.4%), with assigned duties including station maintenance (34.5%), morning PASS checks (34.5%), being a passenger in the fire engine (13.8%), driving the fire engine (9.2%), or driving the EMS vehicle (8.05%). Typical tones and alarms experienced by firefighters included dispatch and heart saver tones, among others. Five or more loud sounds were typically reported on shift by firefighters. Across all shifts recorded among firefighters, 73.7% reported hearing loud noises, though nearly 90% of firefighters did not utilize hearing protection while on shift.

Controls reported listening to music as the highest source of loud noise exposure (46.1%), followed by alarms (19.7%), sirens (17.1%), airplanes (15.8%), traffic (14.5%), restaurants (13.2%), and fitness classes (11.8%) (Table 4). Power tools, children’s toys, sporting events, boats, movie theatres, and lawn mowers were also reported as sources of hazardous noise.

The total duration of loud sound exposure self-reported during a 24 h shift (Table 5, Figure 1) was significantly different between firefighters and controls following Welch’s *t*-test (*t*(73) = −5.3, *p* < 0.001), with the former reporting a median time of 13 min (range: 0–111 min) and the latter a median time of 60 min (range: 0–300 min). Additionally, firefighters experienced a steep drop-off in noise levels observed ~18 to 22 h on shift. This likely corresponds with higher activity and call volumes within fire service during normal waking or daytime hours. While there may be individual peaks in noise levels depending upon the call volume during these hours, on average the noise levels reduced during these hours in the shifts tested. Perhaps unsurprisingly, we did not observe a similarly prominent drop-off among controls, who reported experiencing a variety of recreational loud sounds across all hours, including listening to music, attending noisy restaurants or sports events, and fitness classes, among others. This is also reflected in the larger variability across controls. The maximum dBA recorded on Apple Watches was statistically significant between groups, with firefighters experiencing a median of 87.79 dBA and controls a median of 77.27 dBA (*t*(68) = 6.4, *p* ≤ 0.001). Estimated exposure at 85 dBA, or EET-85 values, was significantly higher for firefighters when compared to controls (Figure 2) (*t*(71) = 8.8, *p* < 0.001). For firefighters, the median EET-85 was 3.97 h (range: 1.20–14.7 h), with 10.81% of shifts exceeding the WHO-recommended range of daily occupational noise exposure. Comparatively, the median EET-85 for control participants was 0.42 h (range: 0.05–8.21 h). Only one control exceeded the recommended the EET-85 guidelines.

Similarly, equivalent exposure 8 h (EA-8hr) was significantly different between firefighters and controls (*t*(71) = 8.8, *p* < 0.001) (Figure 2). The median EA-8hr for firefighters was 81.97 dBA (range: 76.8–87.64) and 72.23 dBA (range: 62.82–85.11 dBA) for controls. In order to better understand how noise exposure varied between groups, we further stratified EA-8hr into three groups, including non-hazardous noise (<70 dBA), tolerable noise (>70 to <85 dBA), and hazardous noise (>85 dBA). Firefighters mostly experienced tolerable total noise exposure (56.76%), though 10.81% experienced hazardous noise exposure above the NIOSH recommended limit. Conversely, controls mostly experienced non-hazardous (31.25%) and tolerable noise (66.67%), with only one individual exposed to hazardous noise (2.08%). Of note, for both groups, the EET-85 and EA-8hr were strongly positively correlated among firefighters and controls (*r* = 1, *n* = 25, *p* < 0.001 and *r* = 1, *n* = 48, *p* < 0.001, respectively).

In order to ascertain the relationship between demographics and EET-85 values, multiple linear regression was also carried out. We examined EET-85 values with firefighter and control cohorts, male and female gender, ethnicity (Hispanic), and age. There was a significant relationship between EET-85 and self-reported non-Hispanic ethnicity (*p* = 0.010), as well with EET-85 and the firefighter cohort (*p* < 0.001). For ethnicity, there was a −0.877 decrease in EET-85 values if individuals identified as non-Hispanic. Additionally, if an individual was a firefighter, EET-85 increased by 1.887. The adjusted R^2^ value was 0.472, indicating that 47% of the variation in EET-85 values can be explained by our model. Multiple linear regression was also conducted for EA-8hr and the demographics described above. We again found that there was a significant relationship between EA-8hr and firefighters (*p* < 0.001) in addition to the EA-8hr and non-Hispanic ethnicity (*p* < 0.001). For the latter, if an individual self-identified as non-Hispanic, their EA-8hr decreased by −3.80. Lastly, EA-8hr increased by 8.1677 in firefighters. The adjusted R^2^ for this model was also 0.472, suggesting that 47% of the variance in EA-8hr can be explained by the model described. For both models, the data met the assumptions of homogeneity of variance and linearity, and the residuals were approximately normally distributed.

## 4. Discussion

In a recent study, we observed significant deficits in cochlear outer hair cell function in the presence of normal audiograms in 176 firefighters [30]. There is an established link between age-related hearing loss and early exposure to loud, hazardous noise. As such, early monitoring and potentially prevention of such exposures presents an important opportunity for improving the long-term health of firefighters and those in occupations with a high-risk of noise exposures. Thus, the primary aim of this work was to characterize hazardous noise exposure in firefighters, a population with known vulnerability to occupational hearing loss. Compared to healthy controls, firefighters had significantly higher EET-85 and EA-8hr values over 24 h shifts, with 10% and 2% of each group exceeding the NIOSH recommended daily limit, respectively. Further, across all controls, only 0.7% of the extracted intervals spanning a 24 h period demonstrated an averaged noise exposure >85 dBA in comparison to 3% of the firefighter intervals. We also found that the types of occupational noise that firefighters are exposed to was fairly homogenous in nature, mostly stemming from sirens, alarms, zone dispatching sounds, and heart saver or defibrillator tones. Further, the variance in max dBA was smaller for firefighters when compared to controls, perhaps suggesting that while the types of noise experienced on shift are loud, there is a maximum level of sound that these alarms and devices emit.

Firefighters were aware of their exposure to hazardous noise. Although none had reported tinnitus or vertigo, they did experience hearing changes following exposure, with 26.7% experiencing tinnitus and 6.7% reporting changes in hearing or muffled hearing. A majority of firefighters also reported experiencing tinnitus, though 48% of controls noted at least rarely experiencing tinnitus as well. Interestingly, despite the above, firefighters were generally not concerned about their hearing following hazardous noise exposure. It is worth nothing that in spite of a lack of compliance with hearing protection while on shift, 60% of firefighters reported using it at least rarely (1–24% of the time) outside of work. Accordingly, 87% of firefighters reported at least slight concern for their hearing later in life. This discrepancy is likely due to the auditory and spatial demands required of individuals employed as career firefighters; in particular, their need to be aware of life-threatening risks while responding to fire and EMS calls.

We also found that there was a significant difference in the duration of self-reported hazardous noise between firefighters and controls, with the latter reporting exposure to noise for up to 300 min. The divergence between groups can be explained by two possibilities: first, it is possible that firefighters are acclimated to a higher baseline level of noise and are not perceptually aware of loud noises in their environment. Secondly, and a more likely explanation, is that firefighters are exposed to short bursts of significantly hazardous noise. Indeed, a major limitation of the Apple Watch as a noise dosimeter is that the noise data as available through the Health App are averaged over half hour intervals, obfuscating the true minima and maxima. Additional limitations of this study include how hazardous noises were characterized and quantified among controls. The Apple Watches did not track how long individuals remained in noisy areas that they self-reported as hazardous, nor was it possible to standardize noise levels for different environments. Further, in addition to being dictated by the mechanism of exposure, it is difficult, if not impossible, to quantify an individual’s spatial relationship to the source of loud noise, which in turn determines the degree of danger posed.

Despite its inherent limitations, a recent study [25] found that overall, the Apple Watch reliably and accurately recorded sound levels when compared to a professional sound level meter, providing further support for the Apple Watch as a convenient noise dosimeter. Thus, it remains an excellent choice to characterize and quantify hazardous noise exposure due to its calibration, convenience, and availability. Future work should take care to incorporate underlying biological mechanisms that might influence susceptibility to NIHL, which almost certainly arises from a combination of genetic and environmental factors [31]. This work would also benefit from quantification with audiologic and vestibular measurements in a longitudinal study.

## 5. Conclusions

In conclusion, this work demonstrates that firefighters are at significantly higher risk of dangerous noise exposures compared to the general public. There is a significant opportunity to reduce risk (along with others at high risk for occupational hearing loss) and influence overall long-term health in this group. The Apple Watch remains a viable tool for characterizing and quantifying occupational hazardous noise exposure in firefighters.

## Figures and Tables

**Figure 1 ijerph-20-02315-f001:**
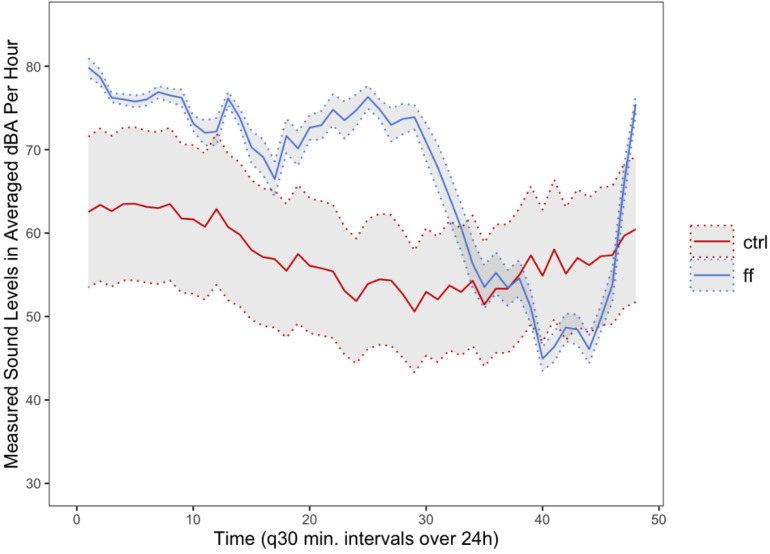
For both groups (firefighters [blue] and controls [red]), the mean was calculated at each interval (every 30 min for each 24 h shift) recorded and extracted from the Apple Watch. The grey cloud is the standard error margin for each time point. Generally, the measured sound levels experienced by firefighters were higher than those experienced by controls with less variability across recorded intervals.

**Figure 2 ijerph-20-02315-f002:**
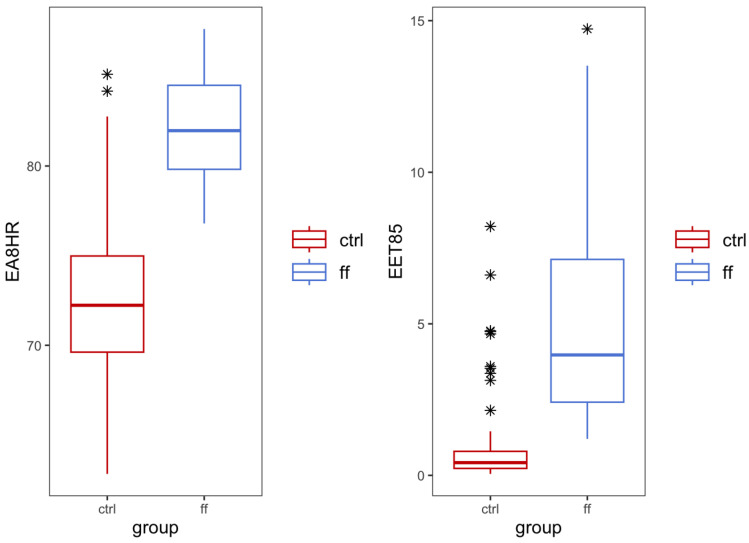
Comparison of EET85 and EA8hr between firefighters and controls. Estimated exposure at 85 dBA (EET85) values was significantly higher for firefighters when compared to controls (*t*(71) = 8.8, *p* < 0.001). Equivalent exposure 8 h (EA8hr) was significantly different between firefighters and controls (*t*(71) = 8.8, *p* < 0.001). * Highlight the outliers.

**Table 1 ijerph-20-02315-t001:** Firefighter and control demographics.

	*Firefighters*	*Controls*
** *n* **	15	25
** *Median Age* **	33 [Range: 23–53]	37 [Range: 21–65]
** *Gender [M/F]* **	12 (80%)/3 (20%)	14 (56%)/11 (44%)
** *Race* **	White (15 [100%])	White (23 [92%]) Asian (2 [8%]) Other (1 [4%])
** *Ethnicity [Hispanic]* **	8 (53.3%)	16 (64%)
** *Education* **	Some college/technical school [66.7%] College graduate [20%] Grade 12/GED [13.3%]	College graduate (18 [72%]) Some college/technical school (5 [20%]) Grade 12/GED (2 [8%])

**Table 2 ijerph-20-02315-t002:** Previous auditory and vestibular symptoms following noise exposure in firefighters and controls.

	*Firefighters*	*Controls*
** *Have you ever experienced any of the following symptoms after a work incident?* **		
*Routine check-up*	3 (20%)	-
*Feeling off-balance*	-	-
*Ringing/buzzing in ears (tinnitus)*	4 (26.7%)	1 (4%)
*Dizziness/vertigo*	-	-
*Changes in hearing*	1 (6.7%)	-
*Muffled hearing*	1 (6.7%)	-
*Other*	2 (13.3%)	1 (4%)
** *Do you hear ringing or buzzing in your ear?* **		
*Never (0% of the time)*	6 (40%)	13 (52%)
*Rarely (1–24% of the time)*	5 (33.3%)	10 (40%)
*Occasionally (25–49% of the time)*	2 (13.3%)	-
*Sometimes (50–74% of the time)*	1 (6.7%)	1 (4%)
*Frequently (75–99% of the time)*	1 (6.7%)	1 (4%)
** *How often do you hear muffled sounds after you are exposed to noise?* **		
*Never (0% of the time)*	9 (60%)	14 (56%)
*Rarely (1–24% of the time)*	3 (20%)	9 (36%)
*Occasionally (25–49% of the time)*	1 (6.7%)	1 (4%)
*Sometimes (50–74% of the time)*	1 (6.7%)	-
*Frequently (75–99% of the time)*	1 (6.7%)	1 (4%)
** *How often do you experience imbalance after hazardous noise exposure?* **		
*Never (0% of the time)*	13 (86.7%)	23 (92%)
*Rarely (1–24% of the time)*	1 (6.7%)	1 (4%)
*Occasionally (25–49% of the time)*	-	1 (4%)
*Sometimes (50–74% of the time)*	-	-
*Frequently (75–99% of the time)*	1 (6.7%)	-
** *How often do you feel disoriented after hazardous noise exposure?* **		
*Never (0% of the time)*	13 (86.7%)	23 (92%)
*Rarely (1–24% of the time)*	1 (6.7%)	2 (8%)
*Occasionally (25–49% of the time)*	-	-
*Sometimes (50–74% of the time)*	1 (6.7%)	-
*Frequently (75–99% of the time)*	-	-
** *How often do you feel you are exposed to hazardous noise?* **		
*Never (0% of the time)*	-	20 (80%)
*Rarely (1–24% of the time)*	3 (20%)	4 (16%)
*Occasionally (25–49% of the time)*	9 (60%)	-
*Sometimes (50–74% of the time)*	3 (20%)	-
*Frequently (75–99% of the time)*	-	-
** *How often do you wear hearing protection during hazardous noise on shift?* **		
*Never (0% of the time)*	6 (40%)	22 (88%)
*Rarely (1–24% of the time)*	4 (26.7%)	-
*Occasionally (25–49% of the time)*	3 (20%)	-
*Sometimes (50–74% of the time)*	1 (6.7%)	-
*Frequently (75–99% of the time)*	1 (6.7%)	-
** *How often do you wear hearing protection during hazardous noise outside work?* **		
*Never (0% of the time)*	7 (46.7%)	18 (72%)
*Rarely (1–24% of the time)*	2 (13.3%)	2 (8%)
*Occasionally (25–49% of the time)*	2 (13.3%)	1 (4%)
*Sometimes (50–74% of the time)*	-	1 (4%)
*Frequently (75–99% of the time)*	4 (26.7%)	1 (4%)

**Table 3 ijerph-20-02315-t003:** Self-reported degree of concern for auditory and vestibular complaints in firefighters and controls.

	*Firefighters*	*Controls*
** *Self-assessed degree of concern for change in hearing after noise exposure* **		
*Not concerned*	4 (26.7%)	16 (64%)
*Slightly concerned*	5 (33.3%)	4 (16%)
*Moderately concerned*	5 (33.3%)	5 (20%)
*Extremely concerned*	1 (6.7%)	-
** *Self-assessed degree of concern for tinnitus after noise exposure* **		
*Not concerned*	5 (33.3%)	16 (64%)
*Slightly concerned*	2 (13.3%)	6 (24%)
*Moderately concerned*	5 (33.3%)	3 (12%)
*Extremely concerned*	2 (13.3%)	-
** *Self-assessed degree of concern for muffled/reduced hearing after noise exposure* **		
*Not concerned*	4 (26.7%)	15 (60%)
*Slightly concerned*	4 (26.7%)	6 (24%)
*Moderately concerned*	6 (40%)	4 (16%)
*Extremely concerned*	1 (6.7%)	-
** *Self-assessed degree of concern for hearing loss later in life* **		
*Not concerned*	2 (13.3%)	6 (24%)
*Slightly concerned*	4 (26.7%)	10 (40%)
*Somewhat concerned*	1 (6.7%)	6 (24%)
*Moderately concerned*	4 (26.7%)	3 (12%)
*Extremely concerned*	4 (26.7%)	-
** *Self-assessed degree of concern for sense of balance?* **		
*Not concerned*	10 (66.7%)	17 (68%)
*Slightly concerned*	2 (13.3%)	6 (24%)
*Somewhat concerned*	1 (6.7%)	1 (4%)
*Moderately concerned*	1 (6.7%)	1 (4%)
*Extremely concerned*	1 (6.7%)	-

**Table 4 ijerph-20-02315-t004:** Firefighter cohort characteristics.

*Firefighters*	*Frequency*	*Controls*	*Frequency*
** *Rank ^a^* **			
*FF/Paramedic/EMT*	10 (66.7%)	-	-
*Firefighter*	4 (26.7%)	-	-
*Driver Operator*	4 (26.7%)	-	-
*Lieutenant*	3 (20%)	-	-
*Inspector*	1 (6.67%)	-	-
*Captain*	1 (6.67%)	-	-
** *Years Employed* **			
*Mean*	11.6 years	-	-
*Median*	6.0 years	-	-
*Range*	0.5–32 years	-	-
** *Typical Unit Assigned* **			
*EMS truck*	11 (28.9%)	-	-
*Fire Engine*	20 (52.6%)	-	-
*Other*	7 (18.4%)	-	-
** *Duties Assigned on Shift* ** * ^b^ *		**Loud Sound Exposures ^c^**	
*Station Maintenance*	30 (34.5%)	Music	35 (46.1%)
*Morning PASS*	30 (34.5%)	Alarms	15 (19.7%)
*Passenger in Fire Engine*	12 (13.8%)	Siren	13 (17.1%)
*Driving Fire Engine*	8 (9.2%)	Airplane	12 (15.8%)
*Driving EMS*	7 (8.05%)	Traffic	11 (14.5%)
** *Median Calls on Shift ^b^* **	5 [Range: 0–9]	Restaurant	10 (13.2%)
** *Median EMS Calls on Shift ^b^* **	2 [Range: 0–9]	Fitness Class	9 (11.8%)
** *Median Fire Calls on Shift ^b^* **	2 [Range: 0–7]	Power Tool	4 (5.3%)
** *Number of Tones on Shift ^b^* **		Children’s Toy	3 (3.9%)
*None*	-	Sporting Event	1 (1.3%)
*1–5 Tones*	8 (21.1%)	Boat	1 (1.3%)
*5–10 Tones*	22 (57.9%)	Movie Theatre	1 (1.3%)
*>10 Tones*	8 (21.1%)	Lawnmower	1 (1.3%)
		Concert	-
		Motorized Sporting Event	-
** *Hearing Protection Utilized* **		**Hearing Protection Utilized**	
*Yes*	4 (10.5%)	Yes	17 (22.4%)
*No*	34 (89.5%)	No	36 (47.4%)

^a^ More than one rank could apply to one individual; participants were asked to select all applicable titles. Participants in both groups were polled at the end of each shift regarding their responsibilities. Frequencies for all totals are from 38 total 24 h shifts recorded in firefighters and 76 total 24 h periods in controls. ^b^ More than one duty may be applicable to one firefighter and ^c^ individuals are likely exposed to more than one type of loud sound. Participants were asked to select all applicable during a shift.

**Table 5 ijerph-20-02315-t005:** Noise metrics for firefighters and controls.

*Noise Metric*	*Median*	*Range*	*% > NIOSH Recommended Limit*
** *Summed Self-Reported Hazardous Noise Exposure* **			
*Firefighters*	13 min	0–111 min	-
*Controls*	60 min	0–300 min	-
** *dBA Minimum* **			
*Firefighters*	37.92 dBA	30.92–48.47 dBA	-
*Controls*	34.77 dBA	30.52–46.33 dBA	-
** *dBA Maximum* **			
*Firefighters*	87.79 dBA	81.28–97.57 dBA	-
*Controls*	77.27 dBA	66.78–83.89 dBA	-
** *Equivalent Exposure Time at 85 dBA (EET-85)* **			
*Firefighters*	3.97 h	1.20–14.73 h	10.81%
*Controls*	0.42 h	0.05–8.21 h	2.08%
** *Estimated Exposure at 8 hr. (EA-8hr)* **			
*Firefighters*	81.97 dBA	76.8–87.64 dBA	10.81%
*Controls*	72.23 dBA	62.82–85.11 dBA	2.08%

The median and range were calculated for summed self-reported hazardous noise exposure, minimum and maximum dBA, and equivalent exposure time at 85 dBA, and estimated exposure time at 8 h (EA-8hr) was categorized as non-hazardous (<70 dBA), tolerable (>70–<85 dBA), and hazardous (>85 dBA). A total of 25 shifts were included for firefighters and a total of 48 shifts were included for controls.

## Data Availability

Derived data supporting the findings of this study are available from the corresponding author (HS) upon request.

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
