# Peer review of "Monitoring Occupational Noise Exposure in Firefighters Using the Apple Watch"

_ijerph, 2023, doi:10.3390/ijerph20032315_

Round 1
Reviewer 1 Report
1. The validity of noise data is not clear. The authors should tell the readers the difference the noise data from the Apple Watch and standard test conditions. The acoustic parameters of the Apple Watch need to be given.
2. The participant numbers in the abstract are different from that in the text.
Author Response
Thank you for the review.
1) Given that Apple Watch is a proprietary device, it is not possible for us to provide technical specifications of the microphone or sound level meter used by Apple. In the revised manuscript, we have included recent references and studies (page 2 and page 9) that have shown that Apple Watch is accurate alternative to traditional sound level meters. We also note that given the nature of the study, it was not possible to use both traditional sound level meter and an Apple Watch with each firefighter for monitoring during their 24-hours service period.
2) We have corrected the participant numbers in the abstract. Thank you for noticing the typing error.
Reviewer 2 Report
In the manuscript ID ijerph-2075439, entitled “Monitoring occupational noise exposure in firefighters using the Apple Watch” the authors aimed to characterize and quantify the noise exposure routinely experienced by firefighters. The authors did not use “traditional methods” for the evaluation of noise exposure, but an Apple Watch for continuous noise monitoring in this group of subject. The work is really well written and structured. The authors properly discuss the results obtained and report the disadvantages and advantages of using the Apple Watch for this kind of studies. I have only one minor comment: I suggest the authors check the abstract for the n of subjects investigated (both firefighters and controls) because I understand that the information reported in the abstract is not with the rest of the text.
Author Response
Thank you for noticing the typing error - we have corrected the participant numbers in the abstract.